Niche conservatism and convergence in birds of three cenocrons in the Mexican Transition Zone

Lizardo Viridiana 1 2
García Trejo Erick Alejandro 3
Morrone Juan J. juanmorrone2001@yahoo.com.mx 1
1 Museum of Zoology ‘Alfonso L. Herrera’, Department of Evolutionary Biology, School of Sciences, Universidad Nacional Autónoma de México , Mexico City , CdMx , México
2 Posgrado en Ciencias Biológicas, Universidad Nacional Autónoma de México , Mexico City , CdMx , México
3 Unit of Informatics for Biodiversity, Department of Evolutionary Biology, School of Sciences, Universidad Nacional Autónoma de México , Mexico City , CdMx , México
Lieberman Bruce
Electronic publication date: 2024 Jan 2
Publication date: 2024
Volume: 12
Electronic Location ID: e16664
Received 2023 Jun 21; Accepted 2023 Nov 21
Copyright: ©2024 Lizardo et al.
Copyright year: 2024
Copyright holder: Lizardo et al.
License: This is an open access article distributed under the terms of the Creative Commons Attribution License, which permits unrestricted use, distribution, reproduction and adaptation in any medium and for any purpose provided that it is properly attributed. For attribution, the original author(s), title, publication source (PeerJ) and either DOI or URL of the article must be cited.
License URL: https://creativecommons.org/licenses/by/4.0/

Keywords: Mexican Transition Zone, Neotropical region, Nearctic region, Evolutionary biogeography, Ecological biogeography, Biotic assembly

Funding: The Consejo Nacional de Humanidades, Ciencia y Tecnologías (CONAHCyT) Viridiana Lizardo was supported by a grant from the Consejo Nacional de Humanidades, Ciencia y Tecnologías (CONAHCyT). The funders had no role in study design, data collection and analysis, decision to publish, or preparation of the manuscript.

==============================
Background

The niche conservatism hypothesis postulates that physiological and phylogenetic factors constrain species distributions, creating richness hotspots with older lineages in ancestral climatic conditions. Conversely, niche convergence occurs when species successfully disperse to novel environments, diversifying and resulting in areas with high phylogenetic clustering and endemism, low diversity, and lower clade age. The Mexican Transition Zone exhibits both patterns as its biotic assembly resulted from successive dispersal events of different biotic elements called cenocrons. We test the hypothesis that biogeographic transitionallity in the area is a product of niche conservatism in the Nearctic and Typical Neotropical cenocrons and niche convergence in the Mountain Mesoamerican cenocron.

Methods

We split the avifauna into three species sets representing cenocrons (sets of taxa that share the same biogeographic history, constituting an identifiable subset within a biota by their common biotic origin and evolutionary history). Then, we correlated richness, endemism, phylogenetic diversity, number of nodes, and crowning age with environmental and topographic variables. These correlations were then compared with the predictions of niche conservatism versus niche convergence. We also detected areas of higher species density in environmental space and interpreted them as an environmental transition zone where birds’ niches converge.

Results

Our findings support the expected predictions on how niches evolved. Nearctic and Typical Neotropical species behaved as predicted by niche conservatism, whereas Mountain Mesoamerican species and the total of species correlations indicated niche convergence. We also detected distinct ecological and evolutionary characteristics of the cenocrons on a macroecological scale and the environmental conditions where the three cenocrons overlap in the Mesoamerican region.

Introduction

Diversity patterns emerge from the overlap of species distributions (Wiens, 2011), which are the geographical representation of ecological niches. The characteristics of these diversity patterns can change since ecological niches can evolve (Holt, 2009) or be conserved over evolutionary time (Wiens & Graham, 2005). According to the niche conservatism hypothesis, physiological constraints prevent species from dispersing into contrasting environments (Wiens & Donoghue, 2004; (Lobo, 2007)). Therefore, it is expected to find richness hotspots, phylogenetically clustered species, and species belonging to older lineages in ancestral climatic conditions (Qian & Ricklefs, 2016). Niche conservatism is a widely described pattern for latitudinal gradients, for example, in mammals (Buckley et al., 2010), birds (Hawkins et al., 2006; Hawkins et al., 2007), insects (Löwenberg-Neto & Carvalho, 2020; Chazot et al., 2021), ferns (Hernández-Rojas et al., 2021), and angiosperms (Qian & Sandel, 2017; Qian, Zhang & Hawkins, 2018; Yue & Li, 2021). There is also evidence of the inverse pattern, known as niche convergence (Qian & Ricklefs, 2016), that emerges from an early dispersal of multiple lineages to novel environments followed by a diversification slowdown (Jablonski, Roy & Valentine, 2006; Qian & Ricklefs, 2016), which causes the co-occurrence of lineages of different ages and backgrounds in environmental conditions that differ from their ancestral niche. This pattern predicts that, as the environmental conditions differ from the ancestral niche, the average clade age increases (Jablonski, Roy & Valentine, 2006; Qian, Zhang & Hawkins, 2018) and phylogenetic clustering decreases (Hawkins et al., 2014; Qian & Ricklefs, 2016; Qian et al., 2019). Niche convergence has been described in plants (Cavender-Bares et al., 2004; Bryant et al., 2008; Cavender-Bares et al., 2011; Culmsee & Leuschner, 2013; González-Caro et al., 2014; Qian, 2014; Hagen et al., 2021), vertebrates (Rolland et al., 2014; Hagen et al., 2021), and Opiliones (Benavides, Pinto-da Rocha & Giribet, 2021).

The Mexican Transition Zone (MTZ) presents an ideal setting to study the interplay between niche evolution and convergence, as it represents an environmentally heterogeneous area where Nearctic and Neotropical lineages coexist. This area has a rich endemic biota distributed in a clear elevational zonation that parallels the latitudinal gradient (Espinosa-Organista et al., 2021; Morrone, 2017), reflecting gradual temperature changes that delimit the Nearctic from the Neotropics (Ficetola, Mazel & Thuiller, 2017). Diversity gradients of the MTZ have been shaped by temperature, latitude, and physiography (Ficetola, Mazel & Thuiller, 2017; Halffter, 2017; Halffter & Morrone, 2017; Morrone, 2020). Evolutionary biotic units called cenocrons, which have an age estimated from various factors, have been used to analyze biotic assembly in the MTZ (Juárez-Barrera et al., 2020). The MTZ was assembled by the successive dispersal events of cenocrons, which are sets of taxa that dispersed from both North and South America (Halffter, 1987; Halffter & Morrone, 2017; Morrone, 2020). Each cenocron represents an evolutionary biotic unit with a distribution linked to current environmental conditions (Lobo, 2007), indicating ‘biogeographic memory’ (Lobo, 1999) or ‘ecological inertia’ (Halffter & Morrone, 2017), which can be interpreted as a measure of niche conservatism.

Each of the cenocrons in the MTZ has a hypothesized minimum age derived from its species distribution size, niche width, phylogenetic affinities, and differential species richness (Juárez-Barrera et al., 2020). These ideas (summarised in Table 1) offer testable hypotheses about the evolutionary processes that led to the biotic assembly of the MTZ. The Nearctic and Typical Neotropical cenocrons have conserved niches due to their recent dispersal, as Wiens & Donoghue (2004) proposed as part of the niche conservatism hypothesis. On the other hand, the Mountain Mesoamerican cenocron has a Neotropical origin but its maximum species richness is distributed in mountain moist forests, which are colder and wetter than the lowlands (their hypothesized ancestral niche). These species also show high endemicity and speciation hotspots at mountaintops as a result of in situ radiations (Halffter & Morrone, 2017; Hagen et al., 2021) and higher diversification rates due to the impossibility of dispersing back to the ancestral niche (Arriaga-Jiménez et al., 2020).

Table 1 Characteristics of three cenocrons that assembled in the Mexican Transition Zone (Halffter, Favila & Arellano, 1995; Halffter & Morrone, 2017).

		Elevation
(m asl)	Dispersal age (ma)	Origin	Niche predictions	
Cenocron	Distribution				Key environmental factors	Niche conservatism	
Mountain Mesoamerican	pine-oak forests, and rain forests in mountains.	1000–2500	37–20	SA	Narrow
(Humidity)	No	
Nearctic	pine-oak forests and mountain grasslands	2500–4000	20–7	NA	Narrow
Temperature (cold)	Yes	
Typical Neotropical	lowlands in the ocean facing side of the Mexican mountain systems and Neotropics. rain forest and deciduous forests	0–2000	7–3.6	SA	Narrow
Temperature (warm)	Yes	
Notes.

NA North American

SA South American

While the cenocron patterns have been described in numerous works (listed in Morrone, 2020), their relationship with niche evolution remains unexplored. Therefore, we aim to test whether cenocrons are composed of clades pre-adapted to local conditions or convergent assemblages with evolutionarily independent lineages, by comparing how clade age, phylogenetic diversity, and species richness correlate with environmental variables. We hypothesize that biogeographic transitionality is a product of niche conservatism in the Nearctic and Typical Neotropical cenocrons, whereas endemicity is a product of niche convergence caused by older dispersal from the Neotropics to colder environments in mountain moist forests and in situ diversification of the Mountain Mesoamerican cenocron (Halffter & Morrone, 2017; Hagen et al., 2021). We expect high and contrasting correlations of the Nearctic and Typical Neotropical species with environmental variables because they have conserved niches. If the Mountain Mesoamerican niches are conserved, the correlations will be comparable to the Typical Neotropical species, whereas if correlations are weak or like those of the Nearctic species, niches will be convergent. To evaluate this, we calculate the richness, phylogenetic diversity, weighted endemism, number of nodes, and age of each cenocron and its correlation with environmental and topographic variables.

Materials & Methods

Study area

The MTZ includes the mountain areas of Mexico, Guatemala, Honduras, and El Salvador (Halffter, 1987; Halffter & Morrone, 2017; Morrone, 2020; De Mendonça & Ebach, 2020). We used a polygon of the MTZ (Fig. 1, left) that includes the Sierra Madre Occidental, Sierra Madre Oriental, Trans-Mexican Volcanic Belt, Sierra Madre del Sur, and Chiapas Highlands biogeographic provinces (Morrone, 2020), and adjacent lowlands. In the mountain bird database, Quintero & Jetz (2018) named this area as the Mexico and Sierra Madre de Chiapas mountain ranges.

Figure 1 The Mexican Transition Zone (Morrone, 2020) and mountain ranges considered from the Quintero & Jetz (2018) database.

Geographic location (left) and environmental space (right) of the Mexican Transition Zone, the Nearctic, and the Neotropics.

To understand the environmental conditions within this area (also called environmental space), we performed a principal component analysis to the bioclim layers from WorldClim 2.0 (Fick & Hijmans, 2017) with a resolution of 2.5 arc min. The details of the analysis and results of the PCA are available in Appendix 1. The first two components explain 80% of the environmental variation (56.8%, and 23.2%, respectively, Table 2) and were used to visualise the existing climate conditions in the MTZ (Fig. 1, right).

Table 2 Importance of components of principal components analysis of mexican transition zone environmental variables.

	PC1	PC2	PC3	PC4	PC5	PC6	PC7	PC8	
Standard deviation	3.016	1.926	1.249	0.85	0.739	0.462	0.26	0.201	
Proportion of variance	56.90%	23.20%	9.70%	4.50%	3.40%	1.30%	0.40%	0.30%	
Cumulative proportion	56.90%	80.00%	89.80%	94.30%	97.70%	99.10%	99.50%	99.70%	

Phylogenetic data

To include the relationships among the bird species for the species selection and phylogenetic diversity measures we used Jetz et al.’s (2012) dated phylogeny with Hackett et al.’s (2008) backbone. The original phylogeny includes all the 9,993 species described by the time of its publication (available at https://birdtree.org/downloads). We used the last set of stage two trees with Hackett’s backbone, which includes 1,000 trees to incorporate phylogenetic uncertainty. We pruned the phylogeny to keep only the selected species, listed in Appendix 2 and plotted in Fig. S1. All the analyses were run through the 1,000 trees set to incorporate phylogenetic uncertainty.

Species selection

The process of species selection involved several steps that resulted in a total of 10 orders, 47 families, 326 genera and 808 species (see Supplementary Material Appendix 2). Most of these species are distributed in the MTZ and Mesoamerica (286 species) or have more than 80% of their distribution in the MTZ (246 species). We removed the following species: (1) those of the aquatic bird orders (Suliformes, Phaethontiformes, Procellariiformes, Charadriiformes, Anseriformes, Podicipediformes, Gaviiformes, Ciconiiformes, Pelecaniformes, Gruiformes, and Eurypygiformes), for not being representative of the Mexican avifauna; (2) migratory families (Parulidae, Hirundinidae, and Motacillidae), since they have an additional complexity due to the niche-switching and niche tracking phenomena (Nakazawa et al., 2004); (3) widespread families (Accipitridae, Cathartidae, Columbidae, Falconidae, Tytonidae and Strigidae), because wide-ranging species drive the variation in richness or average species attribute values (Quintero & Jetz, 2018); and (4) families not distributed in the Americas (Alaudidae, Certhiidae, Otididae, and Prunellidae), for not being relevant for this work.

Then, we selected species listed in the MTZ mountain ranges (Mexico and the Sierra Madre de Chiapas mountain ranges, (Quintero & Jetz, 2018) whose distribution is certain and have native distribution (breeding season and resident) according to the Handbook of the Birds of the World (BirdLife International, 2020). Then, we identified the sister species or clades of each of the species distributed in the study area by finding the parental node and its descendants in all the trees to ensure that only monophyletic groups are analysed using Jetz et al.’s (2012) dated phylogeny with Hackett et al. (2008). All the analyses were repeated through the 1,000 trees included in the set to incorporate phylogenetic uncertainty. Finally, we pruned the phylogeny to keep only the selected species, listed in Appendix 2 and plotted in Fig. S1.

Cenocron assignment

The original concept of cenocron was based on the differential richness of clades as a proxy of the ancestral area (Juárez-Barrera et al., 2020). So, we used Morrone’s (2014) regionalization for the Americas and the ecoregions’ realms for the rest of the world, recoding these areas as the northern hemisphere (Nearctic, Palearctic, or Saharo-Arabian), the southern hemisphere (Afrotropics or Neotropics), or Mesoamerican (Mexican Transition Zone and Mesoamerican dominion). To indicate shared evolutionary history, we assigned each family to a cenocron using these rules: (1) Nearctic if the family has >80% of its species distributed in the Northern hemisphere or if it has more species in the Northern hemisphere than in the Southern hemisphere or Mesoamerica, and phylogenetically related species are Nearctic; (2) Mountain Mesoamerican if a family has more species in Mesoamerica than in the Southern and Northern Hemispheres, and if phylogenetically related species are Neotropical; and (3) Typical Neotropical, if a family has >80% of its species distributed in South America and phylogenetically related species are Neotropical (Fig. 2). A detailed description and code of this section are included in Appendix 1.

Figure 2 Cenocron membership of bird families in the Mexican Transition Zone, based on the number of species distributed in larger biogeographic regions and phylogenetic relationships.

Collection, cleaning, and thinning of occurrence data

We obtained the occurrence data for the years 1950 to 2020 from CONABIO (2021) and GBIF.org (2021a); GBIF.org (2021b); GBIF.org (2021c); GBIF.org (2021d); GBIF.org (2021e); GBIF.org (2021f); GBIF.org (2021g); GBIF.org (2021h). The occurrence query in GBIF included native occurrence data from specimens and observations, marked as without spatial issues, with less than 10 km of uncertainty and a coordinate precision >0.01. Since the data set was too large, it was split into several data sets that can be found on GBIF.org (2021a); GBIF.org (2021b); GBIF.org (2021c); GBIF.org (2021d); GBIF.org (2021e); GBIF.org (2021f); GBIF.org (2021g); GBIF.org (2021h). The database was cleaned using the CoordinateCleaner R package (Zizka et al., 2019) to automatically remove duplicated records, records at sea, and near geographic centroids. We also omitted records outside known breeding distribution according to the Handbook of the Birds of the World (BirdLife International, 2020). Species with more than 5,000 records were down-sampled to 5,000 records and then thinned the data to ensure >5 km of distance between records using the spThin R package (Aiello-Lammens et al., 2015). This distance ensures a minimum separation of two cells between records in the 2.5 arc-minute raster layers utilized in our modelling, minimizing the potential for record clustering and avoiding overfitting. To construct the model, we split the data into training (75%) and testing (25%) for the model construction and validation.

Environmental data and calibration area

We included both ecological and historical conditions to build species distribution models. As ecological factors, we selected variables of high biological importance from WorldClim 2.0 (Fick & Hijmans, 2017) and ENVIREM (Title & Bemmels, 2018). The selected variables fall into three categories: (1) Topographic: elevation, slope, and topographic roughness index, the elevation data from WorldClim were used to create the slope and Terrain Ruggedness Index; (2) Moisture: Thornthwaite’s aridity index, climatic moisture index, and annual precipitation; and (3) Temperature: maximum temperature of the coldest month, minimum temperature of the coldest month, the maximum temperature of the warmest month, minimum temperature of the warmest month. We used a resolution of 2.5 arc min, which was the optimal suitable resolution for conducting a comprehensive macroecological analysis within an environmentally heterogeneous region, given the available computational resources. Each species was modelled with a subset of these variables assuring that the correlation between them was less than 0.5.

To include historical factors, climate layers were cropped to include only those areas that have been accessible to each species over a relevant time (Soberón & Nakamura, 2009). We obtained this area for each species, known as the accessible area or M (Barve et al., 2011), by masking the raster environmental data with the breeding occurrence records and the biogeographic provinces of the Nearctic region (Escalante, Rodríguez-Tapia & Morrone, 2021) and the Neotropical region (Morrone, 2014; Morrone, 2017; Morrone & Ebach, 2022), then added a 1 km buffer, this extra area is added to avoid edge effects and reduce the risk of biased predictions near the distribution boundaries.

Species distribution modelling and hyperparameter optimization

Species distribution models (SDMs) for each selected bird species were constructed to improve the distributional data predictive value and its integration with spatial analysis. We used the MaxEnt algorithm version 3.4.4 (Phillips, Anderson & Schapire, 2006) with the SDMTune package in R (Vignali et al., 2020). This package allows trainng and tuning the hyperparameters of Maxent in an automated way. For each species, we started with a “default model” (regularization multiplier = 1, feature classes = “lqph”, and 500 iterations). Then, we undertook a data-driven variable selection process to get a unique set for each species using the varSel function, selecting a correlation threshold of 0.5, and using AICc as metric to discriminate between models. This function iterates through all the variables starting from the one with the highest contribution and performs a Jackknife test among the correlated variables to remove those that improve the model when omitted. Five species could not be modelled using these hyperparameter tunning and the default model was used.

Then, we performed a grid search to select the best combination of hyperparameters for the model constructed with the selected variables. The parameters tested were regulator multiplier (from 0.8 to 1.2 by a 0.1 increase) and feature classes (“lq”, “lp”, “qp”, and “lqp”). To select the final model, we chose the top five models according to the delta AICc. Then we evaluated the models using the partial ROC (using ntbox Osorio-Olvera et al., 2020), and the omission rate at 5% and selected the model with the highest pRoc value. Evaluation results are included in Appendix 3.

The predicted presence map was obtained with a “clog log” output. To obtain a binary map, we used a Maximum test sensitivity plus specificity threshold (Liu, Newell & White, 2016), since it produces consistent results when using either presence/absence or presence-only datasets, and between rare and common species. This binary map was used for the subsequent analyses. Furthermore, we confirmed visually and manually that our model outputs matched the known distribution according to the Handbook of the Birds of the World (BirdLife International, 2020). A detailed description and code of this section is included in Appendix 1 and the description of the hyperparameters chosen for each species is included in Appendix 3.

Assemblage measures

We calculated various assemblage measures for each cenocron to understand their differences and correlations with environmental variables using the distribution models to calculate raster data of alpha diversity (species richness), age (Mya), number of nodes, phylogenetic diversity (PD), and weighted endemism. Alpha diversity was obtained as the sum of the binary distribution maps of each species. Age is the mean branch length value of all the species found per cell, and number of nodes is the number of nodes from tip to root of all the species found per cell. We made a community matrix from 50 thousand randomly selected points and cells of 2.5 arc min to create a community matrix for the calculation of phylogenetic diversity and weighted endemism. Phylogenetic diversity was calculated with correction for changes in species richness using the tip shuffle model (Laffan & Crisp, 2003), which accounts for changes in species richness when comparing phylogenetic diversity across different communities that could result in skewed results. It randomizes the tips of the tree while preserving the branching patterns and the PD of each community is recalculated. This results in a distribution of phylogenetic diversity values for each community, considering the effects of species richness standardization. Weighted endemism is species richness inversely weighted by species ranges. Both indices were obtained using the package phyloregion R package (Daru, Karunarathne & Schliep, 2020).

Cenocron assemblage’s response to environmental variables

To compare whether these assemblage measures are different between cenocrons, we used a one-way ANOVA with Tukey’s HSD post hoc test, the effect size was labelled according to Makowski et al. (2023). Then, we evaluated whether the assemblage measures are correlated with the environmental variables used to model the species distribution (elevation, slope, topographic roughness index, Thornthwaite’s aridity index, climatic moisture index, annual precipitation, maximum temperature of the coldest month, minimum temperature of the coldest month, maximum temperature of the warmest month, minimum temperature of the warmest month). We applied a Pearson correlation test and compared the coefficient values between cenocrons to detect whether their assemblage responds to environmental variables and the direction of this response (positive vs. negative), and tests with p < 0.05 were accepted as significant. All the analyses were conducted using the R Statistical language (version 4.1.0; R Core Team, 2021).

Diversity in environmental and geographic space

To explore the overlap of species richness in the geographic space, we obtained the sum of the predicted distribution models by cenocron and the total of species. The overlap was also evaluated in environmental space, described as the two principal components that resulted from a principal component analysis to the bioclim layers from WorldClim 2.0 (Fick & Hijmans, 2017) with a resolution of 2.5 arc min, which explains 80% of the environmental variation (56.8%, and 23.2%, respectively, Fig. 1, right). We extracted the values of PC1 and PC2 of the predicted distribution for each species classified by cenocron. Next, we grouped the data by unique coordinates and obtained the richness of each point in environmental space by cenocron and total species. We also counted the number of cenocrons overlapping in each unique coordinate, and selected those coordinates where the three cenocrons were present. The points within the environmental space are interpreted as the environmental transition zone.

Finally, we projected the environmental transition zone into geographic space. This was done by joining the coordinates in the PCA with their original geographical coordinates from the PCA raster layers to visualize the environmental transition zone in a map and compare it with the MTZ (sensu Morrone, 2020, Fig. 1 left).

Results

The three cenocrons had statistically significant differences in all measures (p <0.001) and the effect of the cenocron grouping in all response variables was statistically significant (p < .001). The effect size was medium on age (F(2, 400192) = 19894, p < .001; η2 = 0.09, 95% CI [0.10, 1.00]), mean number of nodes (F(2, 400625) = 24581.46, p < .001; η2 = 0.11, 95% CI [0.12, 1.00]), and weighted endemism (F(2, 401055) = 25878.64, p < .001; η2 = 0.11, 95% CI [0.12, 1.00]). The effect size is large on PD (F(2, 401839) = 76841.89, p < .001; η2 = 0.28, 95% CI [0.30, 1.00]), and on species richness (F(2, 412558) = 57599.38, p < .001; η2 = 0.22, 95% CI[0.23, 1.00]). The Typical Neotropical cenocron has higher species richness, endemism, and PD than the others (Fig. 3). The Nearctic cenocron has a medium crown age and a medium number of nodes, and the distribution of the measure data leans towards lower values. Additionally, this cenocron has more derived species than the other two cenocrons, with lower richness, PD, and endemism. The Mountain Mesoamerican cenocron has a narrower but more uniform distribution in all measures.

Figure 3 Box plot of the assemblage characteristics of the three cenocrons found in the Mexican Transition Zone.

The letters indicate the groups found in a Tukey HSD test result of a one-way ANOVA.

Almost all correlations were significant (p <0.01, Fig. 4), except for a few cases in the Mesoamerican and Typical Neotropical cenocrons. Nearctic and Typical Neotropical cenocrons’ measures are more correlated to the environment than the values of the Mountain Mesoamerican cenocron and the total of the analysed species (Fig. 4). The Nearctic cenocron measures have the highest coefficient values and are positively correlated with higher latitudes and negatively correlated with annual precipitation and temperature, while the Typical Neotropical cenocron correlation estimates are the opposite. The topographic variables are more correlated with the Mesoamerican cenocron measures than with the other two cenocrons. The Mountain Mesoamerican cenocron’s coefficients are low but have the same direction as those of the Nearctic cenocron. Slope and ruggedness are only correlated with the mountain Mesoamerican cenocron. Despite predictions, the Mountain Mesoamerican cenocron is not correlated with any humidity variable.

Figure 4 Pearson correlation test estimates between environmental variables and assemblage characteristics of the birds belonging to the three cenocrons.

Numbers in light grey indicate nonsignificant correlations (p > 0.5).

Geographic richness patterns show the expected overlap in mountain regions (Figs. 5A–5D). The Typical Neotropical cenocron (Fig. 5E) follows the general pattern in the lowlands of both the Pacific and the Gulf of Mexico (Fig. 5D). Nearctic species have a distinct distribution that follows the Sierra Madre Occidental up to Canada (Fig. 5B). Mountain Mesoamerican species are restricted to mountain regions (Fig. 5A), which act as barriers between the Nearctic and Neotropical regions. The environmental distribution of the Typical Neotropical species shows higher species density in the quadrant than the total species pattern but still has the same approximate distribution. Nearctic species (Fig. 5B) are restricted to the quadrant of high-temperature variability, seasonality, and warmer summer (see Fig. 1). Mountain Mesoamerican species (Fig. 5E) share the same environmental space as the Typical Neotropical species.

Figure 5 Geographic (A–D) and environmental (E–H) distribution of bird species density (number of species on a single pixel / number of species in the cenocron) of the Mexican Transition Zone cenocrons.

The authors created this map using original data collected and analyzed by Viridiana Lizardo, CC-BY 4.0, https://creativecommons.org/licenses/by/4.0/.

The geographic representation of the environmental overlap of the three cenocrons is found in the Mexican and Central American mountain regions, including the highlands and lowlands facing the oceans (Fig. 6). The northern limit gradually disappears between the Sierra Madre Occidental and the Arizona Mountain Forests. The southern limit matches with the northern Andes.

Figure 6 Geographic representation of the environmental overlap (grey) between the three cenocrons of the Mexican Transition Zone, compared to the areas (yellow) that correspond to the Mexican Transition Zone (sensu Morrone, 2020).

The authors created this map using original data collected and analyzed by Viridiana Lizardo, CC-BY 4.0, https://creativecommons.org/licenses/by/4.0/.

Discussion

We found that both niche conservatism and convergence occur in the MTZ when the species are split into cenocrons. Assigning species to a cenocron allowed us to detect statistically significant differences between the groups, not only in terms of their average age, endemism, and richness but also in how these measures change along the environmental gradients. The age, richness, and phylogenetic diversity of the Nearctic and Typical Neotropical cenocrons decrease as species move away from their ancestral niche as predicted by niche conservatism. The Nearctic cenocron is correlated with high latitudes, low annual precipitation, and cold environments, while the Typical Neotropical cenocron exhibits the opposite pattern. Contrastingly, the Mountain Mesoamerican cenocron and the total species have low correlations with environmental variables, which indicates that niche convergence may be shaping these diversity patterns (Qian & Ricklefs, 2016). This result is consistent with the predictions for the cenocrons (Table 1) in which recent dispersal results in niches that are conserved towards the ancestral environment (temperate for Nearctic and warm for Typical Neotropical). Even without accounting for the cenocron splitting, the correlations found for the total species suggest that the MTZ is a region of niche convergence.

Relative niche conservatism has been reported in the MTZ, where conservatism was found at the species level but not at the family level, indicating that vicariance drives speciation and then ecological differences evolve (Peterson, Soberón & Sánchez-Cordero, 1999). This is consistent with the recent dispersal of the Typical Neotropical and Nearctic cenocrons into the MTZ and the more ancient dispersal of the Mountain Mesoamerican cenocron. Niches will always be conserved to some extent (Wiens & Graham, 2005), but not among all taxa, environmental variables, or timescales (Stigall, 2014). Examples of both niche conservatism and niche evolution exist in the literature, as reviewed by Wiens & Graham (2005), Pearman et al. (2008) and references cited above. Attempting to explain the complexity of diversity distribution solely based on one of these patterns is an oversimplification, especially in a complex area as a transition zone. So, we focus our discussion on niche evolution and the possible processes behind it.

Cenocron assignment

We assigned species to the cenocrons using simple rules. While previous studies assigned groups to cenocrons based on expert knowledge of the MTZ (Halffter’s original works), used multivariate analyses (Lobo, 2007), or manually linked distribution and phylogenetic data (Roig-Juñent et al., 2018), our approach introduces a more systematic and data-driven method. One of the advantages of this approach is its ability to reduce subjectivity associated with expert knowledge-based classification. Instead, it relies on a set of clear, rule-based criteria that makes it reproducible, as it only uses the species distributions and a biogeographical regionalization.

Our study stands out as the first to test cenocron assemblages for distinct ecological (richness, endemism, and PD) and evolutionary (age and number of nodes) characteristics on a macroecological scale, supporting the idea that cenocrons are not arbitrary groupings but genuine ecological and evolutionary biotic units.

Our method may serve as a preliminary approach for studying cenocrons in the MTZ. While it offers several advantages, the classification method still needs to be improved. Researchers can build upon this methodology by incorporating additional data sources, such as strict phylogenetic information, geological events, or diversification rates.

Conserved niches

Niche conservatism predicts higher species richness within environments occupied by species’ ancestors (Wiens & Donoghue, 2004), and lower age and PD within environments that differ from it (Qian & Ricklefs, 2016). This pattern is observed in the Nearctic and Typical Neotropical cenocrons. The higher species richness within these regions is usually explained by the extended period of colonization, enabling the accumulation of richness over time (Fine, 2015; Pyron et al., 2015; Machac, 2020). Nevertheless, these cenocrons were the last to disperse into the MTZ (Miocene-Pliocene, and Pliocene-Pleistocene, respectively, (O’Dea et al., 2016; Morrone, 2020). Therefore, the species had fewer opportunities to shift niches in comparison with the Mountain Mesoamerican. During this time, as summarized by Mastretta-Yanes et al. (2015), there was a global cooling trend followed by climate fluctuations. In consequence, the distributions of temperate to cold-affinity taxa underwent elevational changes. This is a ‘sky-island’ dynamic, that occurs when species track their preferred habitat in response to environmental changes, which is the main response of a species to environmental changes according to the niche conservatism hypothesis (Lobo, 1999; Stigall, 2014) and a driver of climate-induced vicariance (Claramunt & Cracraft, 2015).

It is important to mention that before the Miocene, South America was an island continent, which resulted in a large diversification and high levels of endemism at continental and bioregional scales (Ricklefs, 2002) and a high specialization to tropical environments. After the Panamanian Bridge lifted and allowed the dispersal of the Typical Neotropical cenocron into the MTZ, birds dispersed as described for the well-understood mammalian interchange (Smith & Klicka, 2010) and their dispersal was also asymmetric. Unlike the northern taxa, the avian families in the Neotropics remain almost entirely restricted to that region (Smith & Klicka, 2010; Smith et al., 2012). For instance, Furnariidae represents the largest continental radiation of Neotropical endemics (Villalobos, Pinto-Ledezma & Diniz-Filho, 2020).

Dispersal from north to south has been more frequent in the MTZ. This is not only because the MTZ had a continuous connection with North America throughout its history (Smith & Klicka, 2010), but also because its mountains are higher when coming from the Neotropics (Janzen, 1967). Therefore, mountain passes are physiological, not topographic, barriers to dispersal (Ghalambor, 2006) and act as such only when there is little or no climatic overlap between the lowlands and the mountains. In temperate zones, the warm season in high elevation can be equivalent to the cold season in the lowlands, which allows species to track the preferred niche seasonally and promotes niche conservatism. This does not happen in the tropics where there is not much seasonal variation in temperature. This hypothesis is linked to the suggestion that seasonal migration promoted colonization of the tropics from the north, which is the mechanism that emberizoid passerines (superfamily Emberizoidea) seem to have followed to colonize the tropics from the north temperate region and is consistent with niche conservatism (Winger, Barker & Ree, 2014). As a result, tropical mountains can be generally structured into distinct thermal zones, with relatively little thermal overlap between low and high-elevation sites across seasons (Muñoz & Bodensteiner, 2019).

Convergent niches

Having discussed the evidence for niche conservatism in the MTZ, we now turn our attention to the Mountain Mesoamerican cenocron, which exhibits patterns consistent with niche convergence. The Mountain Mesoamerican cenocron does not follow the pattern expected when niches are conserved, since it does not correlate with the environmental variables. Its species are distributed at the lower and more variable temperature extremes of the Typical Neotropical cenocron’s distribution (quadrant IV in the environmental space), suggesting that the Mountain Mesoamerican cenocron may represent a subset of the Typical Neotropical cenocron in multivariate space.

The Mountain Mesoamerican cenocron, representing the oldest dispersal in our analysis, has been subject to many climatic fluctuations and landscape shifts. Its species dispersed from South America to Central America during the Oligocene and diversified in the mountains. They further extended their range into Mexico during the Pliocene (Rocha-Méndez et al., 2019; Morrone, 2020). This dispersal took place during Earth’s most recent period of sustained global warmth (Dekens, Ravelo & McCarthy, 2007), characterized by warmer temperatures at high latitudes and reduced temperature differences between the Equator and the poles (Fedorov et al., 2013). At that time, mountains where the Mountain Mesoamerican cenocron dispersed were covered by warm-tempered forests (Mastretta-Yanes et al., 2015). Additionally, there was a highland corridor connecting the mountains of Central America and Mexico through the Isthmus of Tehuantepec (Mastretta-Yanes et al., 2015), which played a role in the geographical setting of the Mountain Mesoamerican cenocron’s dispersal (Halffter, Favila & Arellano, 1995; Halffter & Morrone, 2017; Rocha-Méndez et al., 2019; Morrone, 2020). The Mountain Mesoamerican cenocron spread northwards from Panama across the Central American lowlands, and its lineages started to diverge during the Pliocene at 5.8 Ma (Rocha-Méndez et al., 2019). Subsequently, a period when intense ice age cycles began (Dekens, Ravelo & McCarthy, 2007). In response to these cycles, according to the niche conservatism hypothesis, the main response is dispersal (Lobo, 1999; Donoghue, 2008). However, when it is not possible to disperse back, adaptation to novel climatic niches promotes speciation and/or impedes extinction (Cooney, Seddon & Tobias, 2016).

The fragmentation of the once continuous cloud forest due to Pleistocene climatic fluctuations (Luna Vega et al., 1999), the partial loss of the connection along the Isthmus of Tehuantepec (Mastretta-Yanes et al., 2015), and the lift of high stratovolcanoes in the Trans-Mexican Volcanic Belt (Ferrari et al., 2012) promoted semi-permeable barriers to dispersal (Rocha-Méndez et al., 2019). These events fragmented the cloud forest into areas of high endemism (Luna Vega et al., 1999) and great genetic diversity (Rocha-Méndez et al., 2019). A vicariance model of the Mountain Mesoamerican forests and sky island dynamics explain biogeographic patterns (Luna Vega et al., 1999; Sánchez-González, Morrone & Navarro-Sigüenza, 2008; Fjeldså, Bowie & Rahbek, 2012; Sosa & Loera, 2017) and intraspecific genetic structure (Rocha-Méndez et al., 2019). Furthermore, the current diversity gradients are related to climate (Sosa & Loera, 2017). This agrees with Peterson, Soberón & Sánchez-Cordero (1999), who noticed the importance of separating geographical and environmental space and adding a temporal scale when evaluating niche conservatism. Overall, in situ endemic radiations associated with mountain uplifts and climatic fluctuations (Hagen et al., 2021) may explain the vicariance pattern in the Mountain Mesoamerican cenocron, along with our results that are consistent with a niche convergence diversification model.

Environmental Transition Zone

The Mexican Transition Zone’s provinces overlap with temperate and tropical conditions (Fig. 1), which, according to the environmental overlap hypothesis (Janzen, 1967), make these mountains a semi-permeable barrier. This agrees with the definition of a transition zone as partial barriers or filters that restrict differentially the distribution of each biotic component (Ferro & Morrone, 2014). The area found in our results corresponds to the MTZ (Morrone, 2020), but goes beyond the Nicaraguan Depression. This region represents the distributional boundary for many bird taxa (Sánchez-Ramos et al., 2018), and is considered the southern limit of the MTZ (Halffter & Morrone, 2017). The simpler explanation for this could be the greater dispersal abilities of birds. Nevertheless, the environmental overlap model suggests that the MTZ has a larger overlap with the Neotropics, making it easier for populations to disperse upwards in the mountains or northwards.

Implications for conservation

Conservation efforts should aim to create and preserve corridors that allow species to move and adapt to changing conditions. The environmental transition zone we have described is the place where species from various evolutionary backgrounds coexist, face changing climates by tracking their preferred niches, and encounter isolated habitats that may foster speciation. This underscores the vital importance of preserving this region.

The MTZ is not just a biodiversity hotspot; its diverse species and ecological niches might offer resilience against climate change. Furthermore, the MTZ serves as a valuable natural laboratory for studying ecological and evolutionary processes (Morrone, 2020). Research conducted in the MTZ has the potential to enhance global biodiversity understanding and guide conservation strategies especially when considering the historical context. For instance, Mexican conservation strategies should consider the protection of Neotropical, Nearctic, and species originated in situ in the MTZ.

Study limitations

While our study provides new insights on how to study MTZ’s diversity patterns, we must recognize its limitations. The cenocron concept, which we consider crucial for studying the MTZ, might not be well-known among the broader biogeography and macroecology community. Our study introduces categorization criteria for cenocrons, to make them more accessible and understandable. Yet, there is still potential for improving the methods keeping in mind the importance of incorporating both evolutionary relationships (phylogenetic factors) and environmental influences (ecological factors). Alternative ways to do so are spatial phylogenetics (Mishler, 2023) or the comparative phylogenetic method applied to niche evolution (Evans et al., 2009).

A special challenge in our study was choosing Species Distribution Models (SDM) as data input. since the modeling of species distributions is a complex task with no simple solutions. There are many variations in the way of constructing a model (e.g., data cleaning, calibration area choosing, variable selection or hyperparameter tunning), common problems (like overparameterization), and multiple ways to evaluate the result. Fortunately, there are automatized methods, like the one we used, that allow to perform data-driven variable selection and enable the automation of model hyperparameter tuning. In essence, SDMs are valuable tools for overcoming the Wallacean shortfall, which refers to the incomplete knowledge of species distributions (Bini et al., 2006). SDMs provide the means to compare species with varying data quality and quantity, thus preventing the overrepresentation of species boasting thousands of records while accommodating those with more limited data.

Our primary objective was to understand the broader impact of the environment on diversity within the MTZ. Although not trivial, the methods to model each species will not affect noticeably the results since gradients like the ones described herein have been identified using many data types (points, polygons, lines, grids, transects). Yet, the use of SDM aims to enhance the integration of analyses with spatial methods, and the increased resolution facilitates the identification of gradients in environmentally heterogeneous areas, like the MTZ.

Conclusions

Our analysis supports the expected predictions on niche evolution of the cenocrons in the MTZ. The cenocrons that recently dispersed (Nearctic and Typical Neotropical) show niche conservatism since they are correlated to the measured environmental variables. Contrastingly, the Mountain Mesoamerican cenocron exhibits niche convergence as it shows little to no correlation with the environmental variables. The total avifauna in the MTZ also follows a pattern of niche convergence, which is expected in a biogeographical transition zone. Diversity patterns in complex biogeographic areas like the MTZ are shaped by time-dependent processes that affect species differently. Therefore, it is important to incorporate a historical perspective into ecological studies to better understand these patterns.

Our study shed light on the processes shaping diversity patterns within the MTZ. The Nearctic and Typical Neotropical cenocrons, representing more recent dispersal events, exhibit niche conservatism, with their age, richness, and phylogenetic diversity decreasing as they move away from their ancestral niche. In contrast, the Mountain Mesoamerican cenocron, reflecting the oldest dispersal in our analysis, shows niche convergence, with species distributed at lower and more variable temperature extremes of the Typical Neotropical cenocron’s distribution.

While niche conservatism may drive speciation in some clades, it does not explain overall patterns of diversification (Cooney, Seddon & Tobias, 2016). Recently dispersed cenocrons (Nearctic and Typical Neotropical) tracked the most favourable conditions along the latitudinal and environmental gradient, which allowed them to retain their ancestral niche. In contrast, the Mountain Mesoamerican cenocron, facing multiple environmental changes in a heterogeneous topography, encountered different closest analogues of the ancestral niche for each isolated population (Pyron et al., 2015), leading to the geographical convergence of lineages from different ancestral niches in these mountains.

Our findings emphasize the complexity of biogeographic patterns, which result from intricate evolutionary processes acting at different time scales. Incorporating the historical context, as done by dividing the analysis into cenocrons, is an effective way to incorporate a time-scale perspective since niche conservatism is scale-dependent (Losos, 2008). By doing so, we allow to add historical context to the ecological analyses, which is fundamental to understanding niche evolution.

Supplemental Information

Supplemental Information 1 Full description of data sources, methods, code, and supplementary result data

Click here for additional data file.

Supplemental Information 2 Species used for the analysis, their main distribution, and percentage of their distribution in the MTZ

Click here for additional data file.

Supplemental Information 3 MTZ’s birds SDM Evaluation

Species in alphabetical order, grey indicates that one index is ¡ 0.5.

Click here for additional data file.

Supplemental Information 4 Phylogenetic dataset used, modified from Jetz et al. (2012)

It includes 1000 trees with the selected species.

Click here for additional data file.

This paper is part of V.L.’s doctoral thesis in the Posgrado de Ciencias Biológicas of the Universidad Nacional Autónoma de México (UNAM). The authors thank Federico Escobar and Enrique Martínez-Meyer for their insight and feedback during the planning of this research. Furthermore, we would like to express our gratitude to A. Town Peterson, Bruce Lieberman and the anonymous reviewers for their feedback which significantly improved the quality and clarity of our work.

Additional Information and Declarations

Competing Interests

Author Contributions

Data Availability

Juan J. Morrone is an Academic Editor for PeerJ.

Viridiana Lizardo conceived and designed the experiments, performed the experiments, analyzed the data, prepared figures and/or tables, authored or reviewed drafts of the article, and approved the final draft.

Erick Alejandro García Trejo conceived and designed the experiments, analyzed the data, authored or reviewed drafts of the article, and approved the final draft.

Juan J. Morrone conceived and designed the experiments, analyzed the data, authored or reviewed drafts of the article, and approved the final draft.

The following information was supplied regarding data availability:

The data are available at Zenodo: Viridiana Lizardo. (2023). ViridianaLizardo/birds_MTZ: Hiperparameter method (v3.0). Zenodo. https://doi.org/10.5281/zenodo.10094682.

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
