# Peer review of "Niche conservatism and convergence in birds of three cenocrons in the Mexican Transition Zone"

_PeerJ, doi:10.7717/peerj.16664_

## Round 0.1 · original submission · Major Revisions

The paper has the potential to make a valuable contribution but the reviewers have provided a number of very useful comments and suggestions on your manuscript, both in the review comments section and also on the attached manuscript in the case of reviewer #1, that need to be addressed. Please pay careful attention to all of these when making your revision, as the paper will need to go back out to review. Along with your revised manuscript, please provide a version of the manuscript showing the changes tracked, and a detailed letter explaining the changes you made in response to the reviewer comments and also an explanation for any instances where you diverged from the reviewers.

Reviewer 1 ·

Basic reporting

General comments:
The study presented in the manuscript provides a detailed and comprehensive analysis of the patterns of niche conservatism and convergence in bird assemblages within the Mexican Transition Zone (MTZ). The research is well-conducted, and the results offer valuable insights into the ecological and evolutionary processes shaping species diversity in this biogeographically complex region. Overall, the manuscript is well-structured, the methodology is rigorous, and the findings are clearly presented. However, there are a few minor points that should be addressed to enhance the clarity and scientific rigor of the manuscript:

Experimental design

The manuscript presents a valuable contribution to the field of biogeography by examining niche conservatism and convergence in bird assemblages within the MTZ. With some minor revisions and clarifications, the manuscript will be well-positioned for publication. The study's findings provide important insights into the ecological and evolutionary processes shaping species diversity, highlighting the need for incorporating historical context and considering both ecological and evolutionary perspectives in understanding complex biogeographic patterns.

The methods section is well-described and allows for reproducibility. However, some details could be further clarified. For example, it would be helpful to provide specific information on the selection criteria for the species included in the analysis and the rationale behind removing certain species based on their biology and natural history. Additionally, details on the data pre-processing steps, such as how duplicated records were identified and removed, and niche modeling parameters, could be included for transparency.

Validity of the findings

Results: The results are presented clearly and concisely. The statistical analyses and their outcomes are appropriately reported. The figures and tables effectively illustrate the key findings. However, providing effect size estimates (e.g., eta-squared) without their associated confidence intervals might limit the interpretation of the results. It is recommended to include confidence intervals for effect sizes to better assess the magnitude and uncertainty of the observed effects

Discussion: The discussion is well-structured and provides a comprehensive interpretation of the results. The authors effectively link their findings to previous studies and theoretical frameworks. However, it would be beneficial to discuss the potential implications of the research in a broader context, such as the relevance of niche conservatism and convergence patterns for conservation and management strategies in the MTZ. Additionally, limitations of the study, such as potential biases in occurrence data or uncertainties in species distribution modeling, should be acknowledged and discussed

Conclusion: The conclusion provides a concise summary of the study's main findings and their implications. However, it would be helpful to explicitly state the key contributions of the research and highlight its novelty compared to previous studies on the topic.

Additional comments

Minor Comments:
1. Language and Style: The overall language and style of the manuscript are clear and appropriate. However, there are some minor grammatical and typographical errors throughout the text that should be corrected.
2. Clarity and Structure: The manuscript is generally well-structured and easy to follow. To further improve clarity, it is recommended to provide more explicit transitions between sections and subsections, guiding the reader through the flow of the manuscript.
3. Figures and Captions: The figures are informative and effectively support the presented findings. The captions provide adequate descriptions. However, it would be helpful to include more specific labels or legends in the figures to aid in the interpretation of the results.
4. References: The reference list is comprehensive, and the cited literature is relevant to the study. However, there are a few missing or incomplete references in the text that should be addressed for accuracy and completeness.

Annotated reviews are not available for download in order to protect the identity of reviewers who chose to remain anonymous.

·

Basic reporting

I have concerns about the basic hypothesis. What is stated as the hypothesis to be tested was what I had explained to me on my first visit to the region in the late 1980s. So there is not much new in terms of ideas here ... Maybe that is OK, which would make this analysis a confirmation (if it is robust) of an old idea. I guess that that is OK, but it changes the "air" around the manuscript a bit.

Experimental design

I am afraid that the methodology for the niche models developed and used in this manuscript has many flaws and complications, which can and may have biased or influenced results. Just as a few examples, use of ROC AUC and specificity-sensitivity thresholding (i.e., anything that uses "absence" data) are deeply compromised by the fact that we have only presence data available to us, and that the absence data are pretty much invented in these analyses. Also, the way in which the authors took into account "history" in creating calibration areas is quite dated ... see Machado-Stredel et al. (Machado-Stredel, F., M. E. Cobos, and A. T. Peterson. 2021. A simulation-based method for selecting calibration areas for ecological niche models and species distribution models. Frontiers in Biogeography 13:e48814.). Quite compute-intensive, but much more appropriate. Similarly, using VIF approaches to selecting variables has been shown to be less useful than exhaustive approaches (Cobos, M., A. T. Peterson, and D. Jiménez-Garcia. 2019. An exhaustive analysis of heuristic methods for variable selection in ecological niche modeling and species distribution modeling. Ecological Informatics 53:100983.). Myriad such problems pervade the ENM-based analyses, unfortunately. If the authors wish, they may contact me and I can help them to design a better "recipe."

Validity of the findings

It is hard to judge the validity of the findings of the paper when I have such deep doubts about the ENM methodology, which in essence creates the "data" on which the findings are based.

Additional comments

I am chagrined to have to be so negative about this work, as I am fully conscious of how much work the first author clearly put into developing the analyses. If the editors opt to move forward with publication, that is fine, but I must provide what I consider to be an experience-based, carefully pondered review of what the authors have done in developing the manuscript.

Again, I am happy to consult on assembling a better methodology, if the authors have the stamina for such a "second round."

Very best regards, Town Peterson (University of Kansas)

---

## Round 0.2 · Major Revisions

Some major concerns about the paper continue to crop up in the second round of review. Indeed, it is looking like the authors have not fully resolved the issues from the first round of review, or in doing so, other issues have emerged. I am sorry that the news is not that positive. There has also been difficulty in getting the paper reviewed. If for every paper that someone submitted to a journal, they also were willing to do at least 2 reviews for that journal, then this problem likely would be obviated.

Sincerely,

Bruce S. Lieberman

Reviewer 1 ·

Basic reporting

The text should go English language review. There are many passages that are difficult to understand, especially after the first round of reviews done by the authors.
References should be included to support some of their statements in the discussion
Figures can be slightly improved - the suggestion is in the text
Discussion should be further improved. The section included by the authors "Study limitations" reads like an excuse for the mistakes made during data analysis, that could have in fact been avoided, had the analysis been done a bit more thoroughly.

Experimental design

Methods have to be improved, although I'm not sure if the improvement in the description would improve the results in any way. I believe there was a lot of effort applied to create these results, but indeed the models were not very thoroughly produced and I would suggest creating them using a more robust approach to better support the results. - in kuenm, evaluating a series of models with different parameters should not be difficult, as automating the threshold and evaluation of models, especially if they already have all the data arranged by species is already organized.

Validity of the findings

I waited for the first round of review to see the details of the methods, that were not available in the first submission version. Now that I see it, it is evident that improvement of methods applied to create the models would be relevant before carrying on with the discussion and results.

Annotated reviews are not available for download in order to protect the identity of reviewers who chose to remain anonymous.

---

## Round 0.3 · accepted · Accept

The authors have done a very good job responding to the various rounds of review and have worked hard to address comments and make changes. In my view, the paper is ready to be accepted and go on to the next stage of production.

Sincerely,

Bruce S. Lieberman